# Are Genetic Reference Libraries Sufficient for Environmental DNA Metabarcoding of Mekong River Basin Fish?

Christopher L. Jerde [1,*], Andrew R. Mahon [2], Teresa Campbell [3], Mary E. McElroy [4], Kakada Pin [5], Jasmine N. Childress [6], Madeline N. Armstrong [2], Jessica R. Zehnpfennig [2], Suzanne J. Kelson [3], Aaron A. Koning [3], Peng Bun Ngor [5], Vanna Nuon [7,8], Nam So [5,7], Sudeep Chandra [3] and Zeb S. Hogan [3]

1. Marine Science Institute, University of California Santa Barbara, Santa Barbara, CA 93106, USA
2. Department of Biology, Central Michigan University, Mount Pleasant, MI 48859, USA; mahon2a@cmich.edu (A.R.M.); armst4m@cmich.edu (M.N.A.); zehnp1jr@cmich.edu (J.R.Z.)
3. Department of Biology and Global Water Center, University of Nevada, Reno, NV 89557, USA; tcampbs@gmail.com (T.C.); skelson@unr.edu (S.J.K.); akoning@unr.edu (A.A.K.); sudeep@unr.edu (S.C.); zhogan@unr.edu (Z.S.H.)
4. Interdepartmental Graduate Program in Marine Science, University of California, Santa Barbara, CA 93106, USA; mcelroy@ucsb.edu
5. Wonders of the Mekong Project, c/o Inland Fisheries Research and Development Institute, Fisheries Administration, No. 186, Preah Norodom Blvd., Khan Chamcar Morn, P.O. Box 582, Phnom Penh 12300, Cambodia; pin.kakada77@gmail.com (K.P.); pengbun.ngor@gmail.com (P.B.N.); sonam@mrcmekong.org (N.S.)
6. Department of Ecology, Evolution, and Marine Biology, University of California, Santa Barbara, CA 93106, USA; jnchildress@gmail.com
7. Mekong River Commission Secretariat, P.O. Box 6101, 184 Fa Ngoum Road, Unit 18, Vientiane 01000, Laos; vannanuon88@gmail.com
8. Cambodia National Mekong Committee, No. 576, National Road No. 2, Sangkat Chak Angre Krom, Khan Meanchey, Phnom Penh 12300, Cambodia
* Correspondence: cjerde@ucsb.edu

**Abstract:** Environmental DNA (eDNA) metabarcoding approaches to surveillance have great potential for advancing biodiversity monitoring and fisheries management. For eDNA metabarcoding, having a genetic reference sequence identified to fish species is vital to reduce detection errors. Detection errors will increase when there is no reference sequence for a species or when the reference sequence is the same between different species at the same sequenced region of DNA. These errors will be acute in high biodiversity systems like the Mekong River Basin, where many fish species have no reference sequences and many congeners have the same or very similar sequences. Recently developed tools allow for inspection of reference database coverage and the sequence similarity between species. These evaluation tools provide a useful pre-deployment approach to evaluate the breadth of fish species richness potentially detectable using eDNA metabarcoding. Here we combined established species lists for the Mekong River Basin, resulting in a list of 1345 fish species, evaluated the genetic library coverage across 23 peer-reviewed primer pairs, and measured the species specificity for one primer pair across four genera to demonstrate that coverage of genetic reference libraries is but one consideration before deploying an eDNA metabarcoding surveillance program. This analysis identifies many of the eDNA metabarcoding knowledge gaps with the aim of improving the reliability of eDNA metabarcoding applications in the Mekong River Basin. Genetic reference libraries perform best for common and commercially valuable Mekong fishes, while sequence coverage does not exist for many regional endemics, IUCN data deficient, and threatened fishes.

**Keywords:** eDNA; sequencing; species richness; biodiversity

## 1. Introduction

The molecular genetics revolution that started with sequencing to reconcile species identity and relatedness has led to the use of environmental DNA (eDNA) and high-

throughput sequencing with metabarcoding to survey entire communities from a water sample [1]. The eDNA metabarcoding approach [2] has arguably been most successful to date in fish biodiversity studies [3], and when compared to conventional fisheries surveys, has been shown to perform with parity or better at estimating fish species richness in freshwater systems [4]. In many applications, eDNA-based approaches are cost effective compared with conventional fisheries surveys [5], easily deployed across expansive landscapes [6,7], do not require taxonomic expertise for species identification in the field, and can detect rare, elusive, and unexpected species [8] that are sometimes undetectable by conventional approaches. However, eDNA metabarcoding's main weakness is the need to have a genetic reference library that allows for recovered DNA sequences to be matched to known species [9].

New tools are facilitating the evaluation of genetic reference library coverage. The GAPeDNA web interface assesses global genetic database completeness for fishes using the European Nucleotide Archive [10]. Users may choose between freshwater and marine environments, geographic resolution (provinces, ecoregions, world, or basins), and the mitochondrial position and primer pair used for metabarcoding. Fish species lists for each geographical unit are based on a peer-reviewed database [11], and the primer pairs are similarly peer-reviewed. Many eDNA metabarcoding studies to date have performed preliminary exploration of existing databases (i.e., [12]), but the GAPeDNA interface provides an easy and automated approach, works with existing primer pairs and species lists, and supports the development of applied eDNA metabarcoding efforts.

The Mekong River Basin (MRB; Figure 1) faces numerous threats from a growing human population and the resulting increased demand for resources. Regional stressors include dams and associated fragmentation and hydrological changes, fishing pressure, pollution, sand mining, and climate change-related droughts [13–17]. Given its bioecological and socioeconomic value, particularly its extremely high biodiversity and world's most productive inland fisheries [13,18,19], this system would benefit from expanded monitoring schemes that incorporate eDNA-based approaches. However, like many freshwater systems in tropical regions, the MRB remains underrepresented in published eDNA studies [20].

There are currently five published studies targeting eDNA from aquatic macroorganisms regionally. Species-specific qPCR assays have been developed and successfully applied in situ to detect the Mekong giant catfish (*Pangasianodon gigas*) [21] and the clown featherback (*Chitala ornata*) [22]. In the nearby Chao Phraya River Basin, qPCR was also used to survey for the Chiang Mai crocodile newt (*Tylototriton uyenoi*) [23]. To date, there is only one published eDNA metabarcoding study of fish diversity, conducted near the Nam Theun 2 hydropower reservoir in central Lao PDR [24]. Although eDNA metabarcoding detected more fish taxa than three years of surface gillnet surveys (124 vs. 93 species), genetic identifications were limited because a third of local species lacked references in sequence databases. Additionally, even with two eDNA markers (cyt*b* and 12S), the authors were unable to assign 41–45% of returned sequences to species. This comparison demonstrates the unlocked potential of eDNA metabarcoding monitoring for the MRB.

To assess if genetic reference libraries are sufficient for eDNA metabarcoding of fish in the MRB, a species list must be generated for the area and species group(s) of interest. It is then possible to determine if available primers will amplify species specific sequences for species identification. The Tedesco et al. [11] species database used by GAPeDNA (accessed 22 April 2021) identifies 933 unique fish in the MRB, of which 451 species have reference sequences using a 16S marker developed by McInnes et al. [25]. This was the best primer pair of the 23 possible in the GAPeDNA program. However, other fish lists have more species listed as being in the MRB [26], and some critical species, such as *Urogymnus polylepis* (giant freshwater whipray) and *Balantiocheilos ambusticauda* (burnt tail fish) are missing from reference library consideration based on the default species list of GAPeDNA [11]. Additionally, it is possible to improve fish species detection by combining multiple primer sets whose reference libraries can supplement one another [4,27]. Depending on the

geographic region of interest (e.g., MRB, the Tonle Sap Lake ecosystem) and focal species group (e.g., migratory species, threatened species), the selection of different primer sets may provide improved coverage and performance of the eDNA metabarcoding approach.

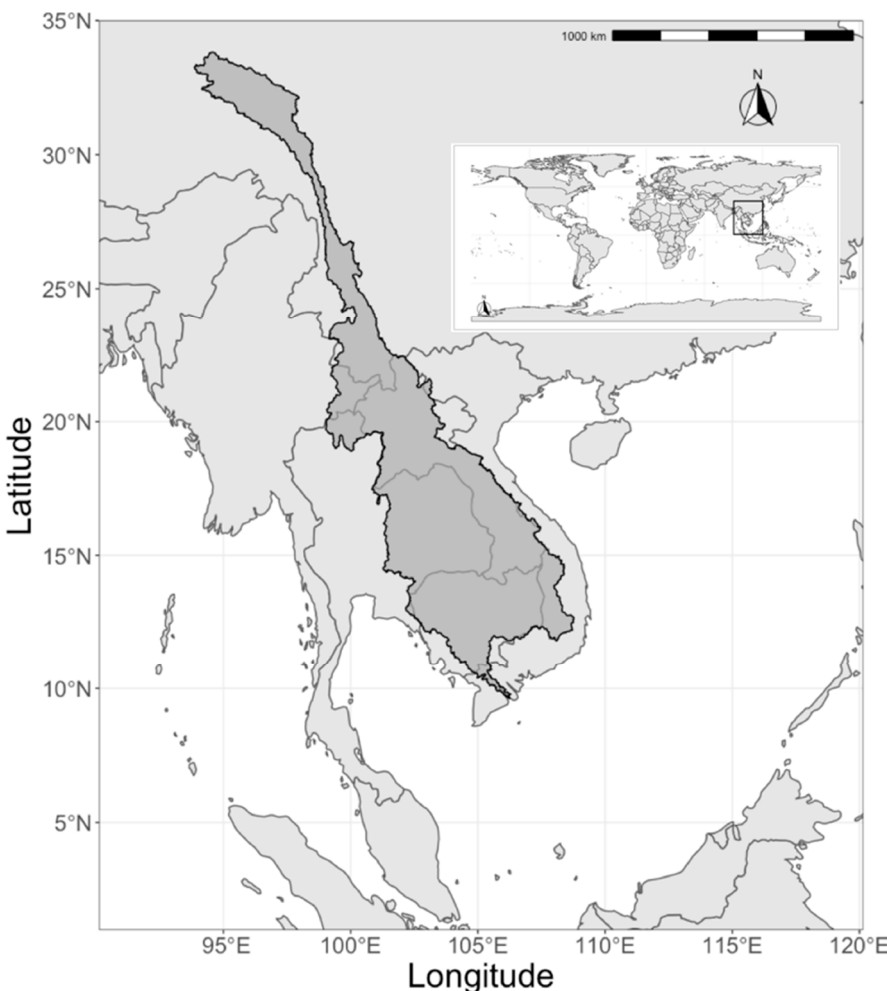

**Figure 1.** The Mekong River Basin (dark gray shaded area) is the longest river in Southeast Asia and flows through six countries: China, Myanmar, Thailand, Lao PDR, Cambodia, and Viet Nam. Black box of inset world map shows the enlarged region of the MRB. The Basin drains an area of 810,000 km² and the large biomass and diversity of freshwater fish recruited and harvested each year partially supports more than 40,000,000 people in the region [13].

Here we describe our in-depth analysis of the genetic references currently available for fish eDNA metabarcoding research in the MRB. We started by compiling four species lists: two lists are considered composites of the MRB, and the other two are presumably subsets of these lists based on political boundaries (Cambodia) and life history (migratory fishes). We then evaluated the genetic reference coverage for each list and identified the primer pairs capable of identifying the most fish. Further, we also assessed a multi-primer pair approach and species specificity within the primer sequences for best performing primer pairs and within critical fish genera for food security and conservation. One of the categorical variables provided by GAPeDNA is the International Union for Conservation of Nature (IUCN) Red List status of each species. For our full species list, we evaluated whether the distribution of IUCN category (e.g., Not Evaluated (NE), Data Deficient (DD), Least Concern (LC), Near Threatened (NT), Vulnerable (VU), Endangered (EN), and Critically Endangered (CR)) is independent of whether at least one sequence is present for a primer pair. Species of conservation and economic value may be disproportionally

overrepresented in genetic libraries. Lastly, we here propose a research agenda for filling key knowledge gaps identified by this analysis to motivate the development of a robust eDNA metabarcoding sampling program in the MRB.

## 2. Materials and Methods

With a length of 4909 km, a watershed of roughly 810,000 km$^2$, and average annual water discharge of 446 km$^3$ year$^{-1}$, the Mekong River is one of the longest and largest rivers in the world [13]. Originating in Tibet at an altitude of about 5200 m, the Mekong flows through China, Myanmar, Laos PDR, Thailand, Cambodia, and Viet Nam. Downstream of China and Myanmar, the river and its associated watershed, is referred to as the Lower Mekong Basin. The Mekong's flood pulse, defined by a maximum wet season discharge 30 or more times the minimum dry season flows, drives ecosystem productivity, which in turn supports one of the largest harvests of freshwater organisms on the planet [13,28–30]. The Mekong's biogeography is notable for distinct patterns of diversity and endemicity throughout the region, with aquatic faunas partly shared between large rivers that once flowed together but now flow apart e.g., the Mekong and Chao Phraya. Together, the size of the river and watershed, elevation change, diversity of habitats, ocean and monsoon influence, immense primary productivity, and geologic history/biogeography have resulted in very high levels of aquatic biodiversity. Moreover, between 1997 and 2007, more than 279 new fish species were named from the basin [31].

We drew upon two databases to delimit MRB fish species. For simplicity, we started with the default species list generated by the GAPeDNA program for freshwater fish (GAP hereafter). The list is sourced from a global database of freshwater fish occurrences by basin and was compiled by extensive searches of available peer reviewed literature, reports, and theses [11]. The second comprehensive fish species list (MRC hereafter) is actively curated by the Mekong River Commission to document the Lower Mekong Basin's rich fish diversity, reconcile species names and identities, and facilitate guidebooks and species lists used to monitor impacts on fisheries by region [13,32].

We supplemented this work with two subsets of the MRB fish lists. The Field Guide to Fishes of the Cambodian Freshwater Bodies [33], represents a geographical subset of Cambodian fish (FCFB hereafter) within Mekong River Basin. In contrast to the GAP but similar to the MRC, the FCFB has marine fish that can occupy fresh and brackish water regularly or periodically, but nevertheless contribute to fish species richness within the MRB. It should be noted that not all of Cambodia falls within the MRB and consequently some fish listed in the FCFB may not occur in the MRB, most notably fishes endemic to southwestern Cambodia including the Cardamom Mountain region. Species lists generated from field guides may be a common starting point for eDNA metabarcoding programs and may document species that are locally known that have not appeared in scientific documents. Because of the ongoing hydropower development of the Mekong River, we also considered a subset of migratory fish species (ZIV hereafter) as defined by Ziv et al. [17]. These two subset databases represent important surveillance programs with different resource management and conservation motivations than only biodiversity monitoring.

For each species list, we attempted to reconcile fish binomial nomenclature synonyms using FishBase [34]. In cases where a fish was listed only to genus without a specific epithet (ex: *Xenentodon* sp.), we removed that listing from further consideration. In circumstances where the genus and species names were provisionally identified with a cf. (ex: *Schistura* cf. *bolavenensis*), we retained the species as the best available identification. When there was any ambiguity or disagreement found in the literature about a particular species identity, we conservatively retained both species names for searching in the genetic databases. Like any fish survey and list, there are likely to be persistent duplication of species based on morphometric description that may ultimately be reconciled with further taxonomic study and genetic sequencing.

Whenever possible, we used the GAPeDNA interface to extract presence or absence of genetic sequences at each of the 23 primer pairs considered. Primer pairs are detailed

in Marques et al. [10] with primer sources [1,5,25,35–48]. For the 933 species listed in GAP, this information was easily compiled and demonstrates the clear advantage of the interface [10]. However, for the remaining species lists (MRC, ZIV, FCFB) we had to customize screening for the presence of sequence data at each primer pair, or in the case of euryhaline or diadromous fishes, we were able to use the GAPeDNA interface for marine species found on the Sunda Shelf. Of the remaining 128 unscreened species using the GAPeDNA interface, 60 had no reference sequences for any region of the mitochondria (or 18S ribosomal DNA).

For the remaining 68 species, available DNA sequence data was manually downloaded from GenBank for all primer pairs in the GAPeDNA system. For each primer pair, downloaded DNA sequences were aligned using MAFFT v7.45 [49]. Primer locations were manually located in BioEdit v7.2.6.1 [50]. For each of the 68 species, primer pairs were included if there was sufficient sequence data to span the forward and reverse primer region. If the species had data that included matching both primers (forward and reverse; data located in regions where primer would bind with sufficient matches visually) and sequence data, it was considered "detectable" and the reference sequences noted as present. Lacking any of this, the sequence would be noted as absent for that primer pair. If there were multiple inconsistencies (nucleotide mismatches) in primer locations, the sequence was considered absent for the marker. The resultant combined species list (UNION hereafter) included 1345 species with presence (1 = yes) or absence (0 = no) of a reference sequence found in the genetic library for each of the 23 primer pairs. The UNION data file with indicator variables for all subset species lists can be found in the Supplementary Materials.

All analyses were conducted in the R program unless otherwise stated. Set theory, that is, the branch of mathematics dealing with defined collections (here species and sequences), informed our analyses of species lists, presence or absence of references sequences, and coverage from single and multiple primers. Hence, we use Euler diagrams to describe both number of species in each list and the overlap in species identity between list proportional to area (R package EulerR) [51]. With respect to primer coverage, we built bar charts for each species list (GAP, MRC, FCFB, ZIV, and UNION), by rank order of species coverage by primer. For the UNION fish species list, we also evaluated the potential to use multiple primer pairs to achieve greater coverage of species by conducting stepwise forward selection.

Closely related species are difficult to differentiate with some of the primer pairs. The MRB provides a unique opportunity by having multiple genera with many species to evaluate this issue. We selected four genera (species within the genus + others), *Pangasius* sp. (13 + 2), *Channa* sp. (9 + 1), *Henicorhynchus* sp. (5), and *Schistura* sp. (75), to explore further. For the *Pangasius* genus we also included *Pangasianodon gigas* and *Pangasianodon hypophthalmus*, as these are closely related to *Pangasius*, presumably have large geographic overlap, are of conservation concern (critically endangered and endangered), and thus are of considerable interest for being differentiated from other species using eDNA approaches. A new record for *Channa auroflammea* is reported in the MRB [52]. This snakehead species is not found in any of our curated lists, but we include it in our analyses of species specificity to evaluate the consequences of new species discoveries on genetic libraries and eDNA metabarcoding approaches.

For each genus in this analysis, sequence data for the mitochondrial region was downloaded from GenBank and aligned using MAFFT v7.45 [49]. After alignment, datasets were cropped to only include data present between the forward and reverse primers of the primer pair having the best coverage. The aligned and cropped datasets were then imported into MEGA-X (i.e., v10) [53]. In MEGA-X, we grouped sequences by identified species and calculated within- and between-species divergence (percent divergence; calculated as uncorrected *p* values). We used a conservative threshold of 5% divergence between species to identify sequence pairings that are unlikely to be distinguished between congeners. We also calculated a measure of within-species sequence variation (>5%) to indicate possible sequence variation due to misidentified uploaded sequences linked to species in GenBank.

Each species in the UNION database has an assigned IUCN status [10], with the default for unassessed species being "Not Evaluated." The other categories are Data Deficient (DD), Least Concern (LC), Near Threatened (NT), Vulnerable (VU), Endangered (EN), Critically Endangered (CR), Extinct in the Wild (EW), and Extinct (EX). We categorized the species into two groups: species having no genetic references and species having at least one sequence in one of the 23 primer pairs. We then evaluated the independence of the groups using a chi-squared test statistic.

## 3. Results

### 3.1. Species Lists

After reconciliation of species name synonyms using Fishbase.org, the MRC species list contained 1135 species listed and the GAP species list had 933. MRC and GAP shared 752 species, but had 383 and 181 unique species, respectively. Unsurprisingly, the majority of species in the FCFB and ZIV lists were also found in one of the MRB-wide lists (MRC or GAP). Of the 29 species found only in the FCFB, they are predominantly euryhaline and/or diadromous fishes that sometimes venture into brackish or freshwater or are found in freshwater systems outside of the MRB [54]. In total, 1345 fish species were considered for primer evaluation under the UNION fish species list representing all species found with the MRC, GAP, FCFB, and ZIV lists (Figure 2).

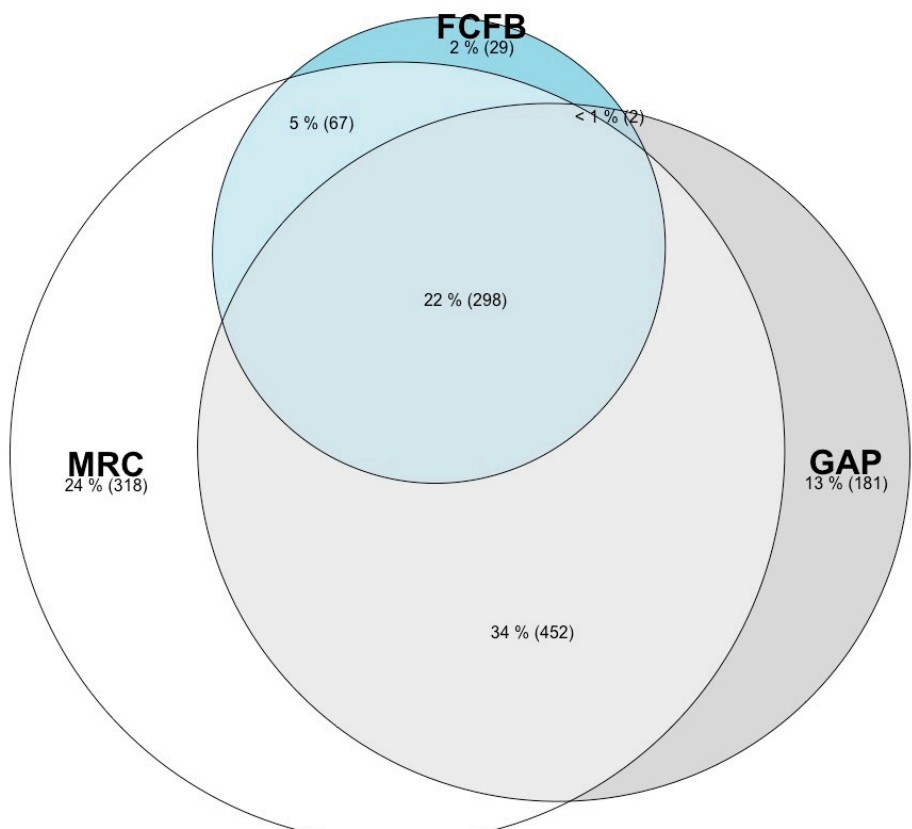

**Figure 2.** Euler diagram of the fish species richness (ellipse size) found within each species list and the shared overlap of species identities. The Mekong River Commission's (MRC; white) list had a total of 1135 species. The GAPeDNA (GAP; gray) list based on Tedesco et al. [11] had 933 species, and the *Field Guide to Fishes of Cambodia Freshwater Bodies* [33] list (FCFB; blue) had 396. The ZIV database (not shown) is comprised of 103 known migratory fish species, 102 of which are found in the MRC database with the remaining one found in the GAP database (100% found within GAP). Percent labels reflect the number identified within the subset divided by the total fish species listed (UNION; *n* = 1345 fish).

*3.2. Single- and Multi-Primer Coverage*

Across all 23 primer pairs, 782 species from the UNION fish list have reference sequences. This represents (782/1345) 58.1% genetic reference library coverage of the fish species in the MRB. Within individual lists, GAP had (545/933) 58.4%, MRC had (661/1135) 58.2%, FCFB had (284/396) 71.7%, and ZIV had (85/103) 82.5% genetic reference library coverage of the fish species. However, this estimate of basin-wide coverage is somewhat deceiving on its own, as many eDNA metabarcoding studies apply only one primer pair [4] due to time and cost considerations, thus severely limiting overall species detections.

Given the fact that most studies can practically use only a limited number of primers, identifying those combinations of available primers that result in accurate identifications for the greatest number of species will provide the highest returns on effort and analytical costs. For example, across our four databases, the top-performing individual primers were the 16S primer pairs put forth by Shaw et al. [45] and McInnes et al. [25]. However, there was nearly identical species identification with both, but the Shaw primer pairs included six additional species in the MRC database not included by using the McInnes primers (Figure 3). Although primers for 18S, CytB, and CO1 regions did not consistently contain sequences for as many species as 16S and 12S across our four fish species lists, they may be critical for identifying species not captured by 16S or 12S. We evaluated the most effective combinations of primers using stepwise forward selection of additional primers to apply. Doing so revealed that of the remaining 22 primers (after having selected the top-performing Shaw 16S primer), a CytB primer (Thomsen cb) added the most new identifications to the list: 80 species. Note that in this instance, top-performing denotes only that a reference sequence is present, but does not consider amplification performance or species specificity.

As the top-performer, the Shaw 16S primer pair captured 643/782 (82.2%) of fish species with a genetic reference in the GenBank library, but only identified 643/1345 (47.8%) of the total basin-wide fish species richness as described by the UNION fish species list. For the remaining species and primers, the CytB Thomsen cb primer added the most species by adding sequences for 80 new species. By iterating this process of maximizing species coverage while using the fewest primers, we found that six primer pairs provided 98.5% coverage of all species having sequences in the genetic reference library and 57.4% coverage of fish in the UNION species list (Table 1). In addition to providing broader species coverage, multiple primer studies add greater potential to differentiate species [55]. Thus, future applications of eDNA studies would likely benefit by considering the species representation offered by these top-performing primer subsets.

**Table 1.** Stepwise selection of primer pairs for UNION fish species list with 782 species with sequences.

| Step | Primer Pair(s) | Species with Sequences | Percent of Species with Sequences (*n* = 782) | Percent of Total Species in UNION (*n* = 1345) |
|---|---|---|---|---|
| Step 1 | 16S Shaw | 643 | 82.1% | 47.8% |
| Step 2 | 16S Shaw CytB Thomsen cb | 723 | 92.5% | 53.8% |
| Step 3 | 16S Shaw CytB Thomsen cb CytB Miya | 745 | 95.3% | 55.4% |
| Step 4 | 16S Shaw CytB Thomsen cb CytB Miya 12S Bylemans | 759 | 97.1% | 56.4% |

**Table 1.** *Cont.*

| Step | Primer Pair(s) | Species with Sequences | Percent of Species with Sequences (*n* = 782) | Percent of Total Species in UNION (*n* = 1345) |
|---|---|---|---|---|
| Step 5 | 16S Shaw CytB Thomsen cb CytB Miya 12S Bylemans 12S Kelly | 766 | 98.0% | 56.9% |
| Step 6 | 16S Shaw CytB Thomsen cb CytB Miya 12S Bylemans 12S Kelly CytB Thomsen 2deg | 772 | 98.7% | 57.4% |

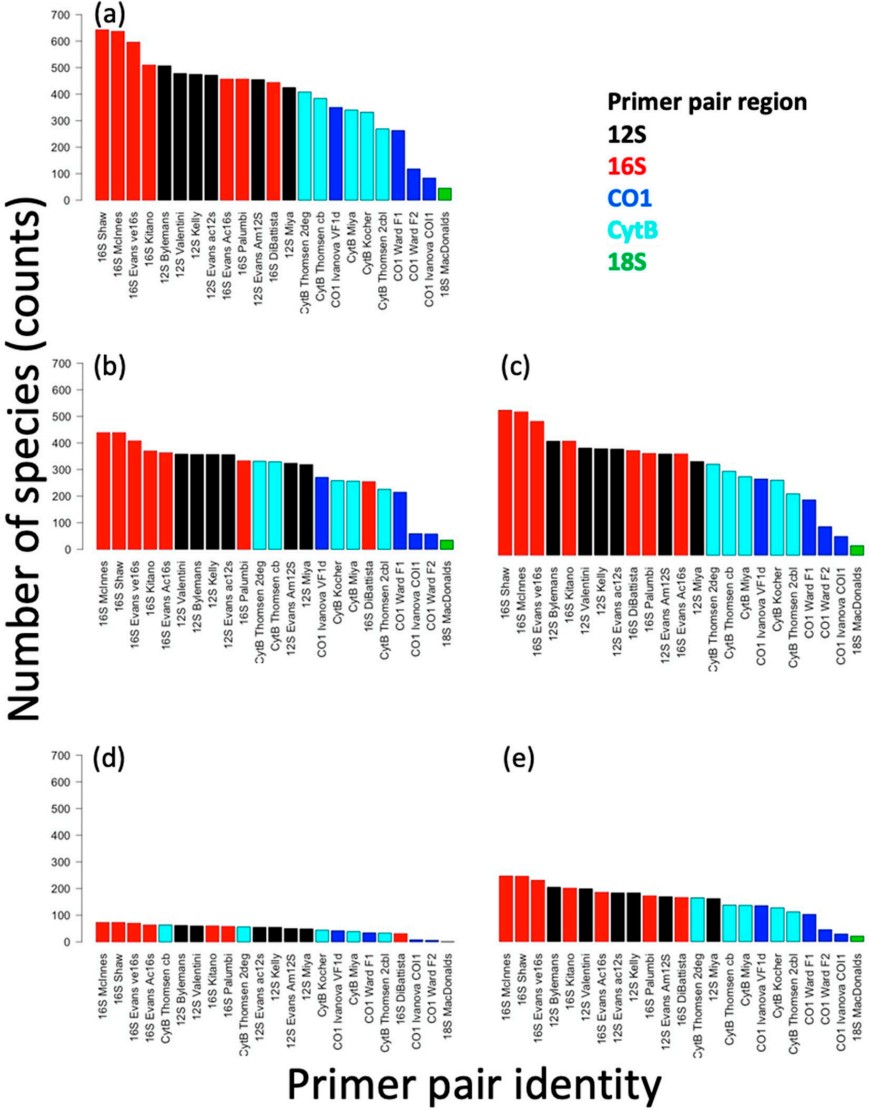

**Figure 3.** Bar charts of primer pair coverage for all species lists in the MRB ((**a**): UNION), using the GAPeDNA default ((**b**): GAP), the curated MRC database ((**c**): MRC) and the two subset databases by migratory fish species ((**d**): ZIV) and by country of Cambodia ((**e**): FCFB). Primer pair names are consistent with GAPeDNA identities [10]. Primer locations are color coded and the 16S Shaw [10,45] and 16S McInnes [10,25] primer pairs are consistently the best performing primer pairs across databases.

### 3.3. Species Specificity

Within the *Channa* genus all species but one had a sequence within the Shaw 16S primer pair. For the one species without a sequence, *C. melanoptera*, there were no sequences across the 22 other primer pairs (Table 2). Three species, *C. marulius, C. melasoma*, and *C. auroflammea* had >5% similarity with each other and are possibly indistinguishable using the Shaw 16S primer pair. In total, there is good evidence that eDNA metabarcoding with the 16S primer region would provide sufficient sequence coverage and specificity to detect seven of the *Channa* sp., assuming adequate amplification of the primer pair.

The *Henicorhynchus* genus showed an opposite result to the *Channa* genus (Table 2). The *Henicorhynchus* genus had five species in the MRB species lists, and while there was good sequence coverage for each of the species across multiple primer pairs, the between species with >0.05 genetic similarity between species indicates that the ability to differentiate between species is unlikely. Other primer pairs (existing or yet to be developed) may provide better discrimination.

Within the *Pangasius* genus, potential problems with species-level specificity originate from outside of the genus. Of the 11 *Pangasius* sp. with Shaw 16S sequences, 10 show the ability to partially match with sequences of *Panagasianodon hypophthalmus*. In contrast to *Henicorhynchus* genus where species within the group are potentially not discernable, this is an instance where species outside the genus may cloud the detection of *Pangasius* sp. Yet the solution is similar–a different primer pair may work better. Alternatively, there may be a problem with misidentified sequences within *Panagasianodon hypophthalmus* uploaded to the genetic databases. With 16% within-species variation, the largest in our study, *P. hypophthalmus*'s genetic identity will need to be verified with voucher specimens for secure inferences.

Despite comprising approximately 5% (77/1345) of the known species in the MRB, the *Schistura* genus has only two sequenced species. There is no way of assessing if other *Schistura* sp. can be detected as *S. fasciolata* or *S. kaysonei*, and as a result, eDNA metabarcoding is essentially blind to the presence of most *Schistura* species irrespective of the primer pair used.

**Table 2.** Evaluation of species specificity (N.E. = Not Estimable).

| Species Identity | Sequence Presence in Shaw 16S (Yes/No) | Within Sequence Similarity (Shaw 16S) | Between Species with >0.05 Genetic Similarity (Shaw 16S) | Number of Primer Pairs with Sequences (23 Max) |
|---|---|---|---|---|
| *Channa* spp. | | | | |
| *C. gachua* | Yes | 6% | None | 18 |
| *C. lucius* | Yes | 4% | None | 19 |
| *C. marulioides* | Yes | N.E. | *C. auroflammea* *C. marulius* | 5 |
| *C. marulius* | Yes | 1% | *C. auroflammea* *C. marulioides* | 20 |
| *C. melanoptera* | No | N.E. | N.E. | 0 |
| *C. melasoma* | Yes | 1% | None | 5 |
| *C. micropeltes* | Yes | 1% | None | 17 |
| *C. orientalis* | Yes | N.E. | None | 5 |
| *C. striata* | Yes | 3% | None | 19 |
| *C. auroflammea* [1] | Yes | N.E. | *C. marulioides* *C. marulius* | N.E. |
| *Henicorhynchus* spp. | | | | |

**Table 2.** *Cont.*

| Species Identity | Sequence Presence in Shaw 16S (Yes/No) | Within Sequence Similarity (Shaw 16S) | Between Species with >0.05 Genetic Similarity (Shaw 16S) | Number of Primer Pairs with Sequences (23 Max) |
|---|---|---|---|---|
| *H. caudimaculatus* | No | N.E. | N.E. | 1 |
| *H. entmema* | Yes | N.E. | N.E. | 15 |
| *H. lineatus* | Yes | 7% | *H. entmema* *H. ornatipinnis* *H. siamensis* | 18 |
| *H. ornatipinnis* | Yes | N.E. | *H. entmema* *H. ornatipinnis* *H. siamensis* | 4 |
| *H. siamensis* | Yes | 0% | *H. entmema* *H. ornatipinnis* *H. siamensis* | 16 |
| *Pangasius* spp. and *Pangasianodon* spp. | | | | |
| *Pangasius bocourti* | Yes | 1% | *P. polyuranodon* *P. macronema* *P. hypophthalmus* | 12 |
| *P. conchophilus* | Yes | 0% | *P. macronema* *P. hypophthalmus* | 10 |
| *P. djambal* | Yes | 0% | *P. macronema* *P. hypophthalmus* | 3 |
| *P. elongatus* | Yes | N.E. | *P. hypophthalmus* | 4 |
| *P. krempfi* | Yes | 0% | *P. macronema* *P. hypophthalmus* | 13 |
| *P. kunyit* | No | N.E. | N.E. | 0 |
| *P. larnaudii* | Yes | 0% | *P. polyuranodon* *P. macronema* *P. hypophthalmus* | 19 |
| *P. macronema* | Yes | 0% | Most *Pangasius* sp. with sequences | 14 |
| *P. mekongensis* | No | N.E. | N.E. | 0 |
| *P. nasutus* | Yes | 1% | *P. hypophthalmus* | 10 |
| *P. pangasius* | Yes | 0% | *P. macronema* *P. hypophthalmus* | 20 |
| *P. polyuranodon* | Yes | N.E. | *P. larnaudii* *P. bocourti* *P. hypophthalmus* | 10 |
| *P. sanitwongsei* | Yes | 1% | *P. macronema* *P. hypophthalmus* | 13 |
| *Panagasianodon gigas* | Yes | 0% | *P. macronema* | 19 |
| *P. hypophthalmus* | Yes | 16% | All *Pangasius* sp. with sequences | 20 |
| *Schistura* spp. | | | | |
| *S. fasciolata* | Yes | 5% | N.E. | 16 |
| *S. kaysonei* | Yes | N.E. | N.E. | 16 |
| +73 *Schistura* spp. | No | N.E. | N.E. | 0 |

[1] Not found in UNION species list.

*3.4. IUCN Status*

In UNION species list, there were 782 species with at least one reference sequence across 23 primer pairs. Of these species, the IUCN designated 154, 81, 466, 29, 29, 13, and 10 species as Not Evaluated (NE), Data Deficient (DD), Least Concern (LC), Near Threatened (NT), Vulnerable (VU), Endangered (EN), and Critically Endangered (CR), respectively. Note that Extinct in the Wild (EW) and Extinct (EX) are excluded from consideration of the species lists following GAPeDNA's default settings. Of the 563 species with no

reference sequences, the IUCN designated 192, 142, 171, 11, 20, 13, and 14 species as NE, DD, LC, NT, VU, EN, and CE, respectively. The chi-square independence test of the contingency table yielded a $x^2$ of 136 with 6 degrees of freedom and a resulting $p$-value < 0.001. The conclusion is that categories are not independent of each other. Notable discrepancies between observed and expected values occurred with the number of LC with at least one primer pair sequence and the number of DD without at least one primer pair sequence. Conclusions were similar for the GAP and MRC species lists (not shown). The FCFB and ZIV species lists were not assessed due to issues with some categories having zero observations, which does not allow for statistical evaluation.

## 4. Discussion

With the easy-to-use web interface, Marques et al. [10] have developed a valuable interface for fish biodiversity and conservation managers who are considering the implementation of an eDNA metabarcoding surveillance program. However, given the 57.1% concordance between the default GAP and MRC fish species lists ((452 + 300)/(1345 − 29)) (Figure 2), the GAPeDNA platform is a useful but incomplete resource for assessing the coverage of genetic reference libraries and identifying species requiring further sequencing. The MRB provides a challenging case study and reveals some of the persistent concerns about implementing eDNA metabarcoding in ecosystems with high fish biodiversity [4,56]. These challenges are not exclusive to the GAPeDNA platform and include assessing discrepancies across place-based species lists, the limited capacity for single-marker approaches to comprehensively monitor fish species richness, the absence of reference sequences for fish species of concern, no species specificity within and between some genera for many primers, and the potential unreliability of species taxonomic identification matched to sequences in reference databases.

New versions of sequence coverage screening software, like GAPeDNA, will ideally allow more flexibility to evaluate customized species lists, particularly if key species are notably absent from default lists. For example, the GAP species list was missing *Cyclocheilichthys armatus, Labeo pierrei,* and *Pangasius mekongensis*, all of which were found in MRC, FCFB, and ZIV with no indication of misidentification due to a name change. These three species are also of conservation concern because of their migratory life history requirements and the potential impacts from dams [17]. The MRC species list is very comprehensive and actively curated whereas Tedesco et al. [11], though published and peer-reviewed, is a static resource. The FCFB also demonstrates a nuance for species richness monitoring where some marine species may contribute to the overall biodiversity in freshwater systems seasonally, but may be precluded from species lists depending on the criteria for inclusion. This is a consideration for other studies where a river basin has a terminus at the ocean, or alternately, marine and brackish water environments that may have freshwater fish species occasionally found in estuaries and deltas [4].

With 1345 species in the UNION species list, we have advocated for inclusiveness in order to facilitate robust fish biodiversity monitoring. However, the list undoubtedly includes species with two or more binomial nomenclatures that have not been genetically evaluated and differentiated. This will inflate the species richness estimate for fishes from the UNION data set. However, this list, with consideration of the other datasets (GAP, MRC, ZIV, and FCFB), serves as an opportunity to identify discrepancies, and because of the motivation to build out eDNA metabarcoding reference libraries for the MRB, can also facilitate genetic evaluations of species, particularly if nearly entire genera appear to be absent from existing databases (i.e., *Schistura* spp.). As a recommendation going forward for using sequence coverage screeners before implementing an eDNA metabarcoding surveillance program, it is potentially advantageous to consider multiple species lists to ensure wide species coverage and identify knowledge gaps where genetic sequencing efforts can be doubly useful in reconciling species and allowing for genetic detection.

The UNION species list also represents the broadest list of fish species presumably found within the MRB. Some conservation research questions will not require such a de-

tailed list. For example, eDNA metabarcoding efforts for the Tonle Sap Lake Ecosystem may have far fewer species as localized endemics from the upper headwaters do not occur there. Migratory species as represented by the ZIV species list are well represented already and could be completely screened and uniquely identified with additional mitochondrial genome sequencing of 18 additional species (85 of 103 species have at least one sequence present in the ZIV database). However, ensuring coverage for any primer pair and species specificity within the primer pair will take effort beyond these 18 additional fish species. Nevertheless, the geographical and conservation scope of the research will be critical for ensuring reliable inferences [57,58].

Even the best performing primer pairs, namely 16S McInnes or 16S Shaw (Figure 3), do not have reference sequences for coverage of even half the MRB fish species (Table 1), and the multiple primer pair approach may be desirable or needed. The multiple primer pair approach, sometimes referred to as using multiple markers, can achieve up to 57% coverage in the MRB, but the cost for sequencing may be prohibitive and further limited by the amount of DNA recovered from a water sample in order to use six primer pairs. Nevertheless, the multiple primer pair approach has been useful for estimating fish species richness [4], especially when there are many species within a genus [55] as congeneric species are more easily differentiated by particular primer pair combinations.

Ultimately, whether using a single or a multiple primer pair approach, genetic coverage alone does not ensure eDNA metabarcoding can reliably survey fish communities to species level. For some genera, such as *Channa* spp., the library coverage is good and discrimination between species appears reliable using the 16S Shaw primer pair. However, even with good coverage of *Henicorhynchus* spp. and *Pangasius* spp., there is considerable uncertainty regarding whether recovered sequences are sufficiently species specific. Indeed, as pointed out by Marques et al. [10], it appears the 12S region of the fish mitochondria, although having low coverage in genetic reference libraries including the MRB, often provides better species specificity. Future sequencing effort in the MRB may emphasize sequencing for 12S specifically, or given the decreases costs for sequencing, the whole mitochondrial genome.

Future metabarcoding efforts may also benefit from additional screening of primer pairs for amplification bias with in silico PCR programs, which is a known phenomenon in eDNA metabarcoding [59]. Amplification biases occur when a primer pair preferentially amplifies DNA from certain taxa and not others, and this can lead to unanticipated false negatives when DNA present in a sample is not amplified and not detected. This is of particular concern with more universal genetic markers like COI and cytochrome *b*, which can amplify a wider range of taxa, than with fish-centric 12S and 16S markers [60,61]. Programs like EcoPCR, PrimerTree, and MFEprimer-2.0 allow practitioners to run 'virtual' PCRs and assess a priori how well a primer pair will amplify DNA from taxa with existing reference sequences [62–64]. Marques et al. [10] used EcoPCR and discovered 4 out of 23 selected primer pairs would only amplify <0.05% of global fish taxa and subsequently excluded these primers from further analyses. In silico programs can help practitioners narrow their primer selection in advance, avoid potential wasted sequencing effort, and evaluate whether PCR bias may account for non-detection of certain taxa.

The difference in genetic library coverage between *Schistura* spp. and the migratory fish species identified by Ziv et al. [17] (ZIV) demonstrates that genetic libraries, and research agendas more broadly, often favor charismatic or commercially valuable species over others. Of the 103 species in the ZIV database, 85 species have some genetic sequencing in at least one of the primer pairs. In contrast, *Schistura* spp. have only two of 75 species with 16S Shaw primer pair coverage (Table 2) and 10 of 75 with genetic sequencing across any of the 23 primer pairs. These stone loach species found throughout southern and eastern Asia are difficult to morphometrically identify to species, and there is very little information to genetically differentiate them in the eDNA metabarcoding gene regions evaluated here, yet there is a growing effort to reconcile phylogeny [65].

The other species specificity issue revealed in our study is somewhat speculative, but is a known problem. Genetic databases, such as GenBank and BOLD, rely on careful taxonomic identification and proper uploading of the sequence information for each species [66]. As exemplified by the *P. hypophthalmus* potentially matching to multiple species of the *Pangasius* genus and the large within-species variability of the *P. hypopthalmus* sequences (16%; Table 2), it is very possible there are multiple misidentified sequences in the reference database. However, genetic reference databases are improving rapidly through improved curation resulting in less than 1% error rate at the genus level [67], but confidence in species-level inferences is wanting and may require targeted efforts to link voucher specimen identification to genetic sequences.

There were, somewhat unexpectedly, a large number of Least Concern (LC) species with some sequence coverage in the UNION species list relative to species without any sequence coverage, and also more Data Deficient (DD) species lacking a reference sequence in the library than expected. This could reflect an absence of research on rare species, those found in hard to access locations, and/or those species not of critical food or high conservation value. There are 58 species listed at Near Threatened, Vulnerable, Endangered, or Critically Endangered that have no reference sequences across any of the 23 primer pairs. This constitutes 4.3% of the total species in the UNION database. There are 192 Not Evaluated and 142 Data Deficient, or approximately 25% of total fish species that have no reference sequences across any of the 23 primer pairs. Presumably some of these species fall into categories of species of concern and the 4.3% value should be seen as an underestimate. Due to the construction of dams throughout the MRB, migratory species are priority targets for sequencing. These species include: *Aaptosyax grypus*, *Acanthopsoides delphax*, *Bangana behri*, *Brachirus harmandi*, *Cirrhinus jullieni*, *Cyclocheilichthys apogon*, *Cyclocheilichthys furcatus*, *Cynoglossus microlepis*, *Hemisilurus mekongensis*, *Himantura krempfi*, *Hypsibarbus lagleri*, *Hypsibarbus pierrei*, *Lobocheilos cryptopogon*, *Osteochilus enneaporos*, *Pangasius kunyit*, *Pangasius mekongensis*, *Paralaubuca harmandi*, and *Probarbus labeamajor*. Given their significance as migratory species of conservation concern, it would be prudent to consider whole genome sequencing of these species for improved primer pair coverage and the ability to differentiate them to species. The lowering cost and technological advancement of genetic sequencing is making it possible for whole genomes to be readily screened. Ultimately having complete fish communities with entire genomes sequenced will lead to better primer pair selection and potentially fewer primers needed for any given surveillance effort.

Similarly, there are many genera without the genetic information to build confidence in eDNA metabarcoding's ability to detect and differentiate species. Examples of genera with species (*n*) having no genetic coverage include A*kysis* (8), *Glyptothorax* (9), *Lobocheilos* (9), *Poropuntius* (13), *Pseudobagarius* (8), and *Schistura* (65). Many, but not all, of these species, as we speculated previously, are not easily identified, caught, nor common food resources.

The MRB is a challenging system for eDNA metabarcoding. And yet, with the aid of GAPeDNA and additional research targeted at improving specificity testing, many fish species could potentially be monitored using this approach. There remains substantial work to be done to make eDNA metabarcoding of fish species effective and reliable, even for subsets such as genera (i.e., *Channa*) or geographic regions (Cambodia). The screening of reference libraries in less diverse systems has been used to calibrate eDNA metabarcoding and there is growing confidence that with careful selection of primer pairs and improved reference libraries the approach can be implemented for active conservation management of entire fish communities [4], but as we found here, assessment of species presence or absence under current eDNA metabarcoding conditions should be made with caution. To answer the title question of this research, "are generic reference libraries sufficient for eDNA metabarcoding of Mekong River Basin fish?"; we can state, not yet. Global fisheries are facing unprecedented challenges and eDNA metabarcoding is emerging as a powerful tool for monitoring environmental change and fisheries dynamics [3]. However, the inferences gained from the eDNA metabarcoding approach are contingent on ensuring the genetic infrastructure is available in the form of populated genetic reference libraries

for species found in diverse systems and primer pairs used that can differentiate species. More work on eDNA metabarcoding is needed in the MRB, and globally, to assess, monitor, and protect freshwater fish species and critical fisheries.

**Supplementary Materials:** The following are available online at https://www.mdpi.com/article/10.3 390/w13131767/s1, UNION.csv: data file used for analysis of sequence coverage.

**Author Contributions:** Conceptualization, C.L.J., A.R.M., and Z.S.H.; methodology, C.L.J., A.R.M., T.C., and Z.S.H.; formal analysis, C.L.J., M.N.A., J.R.Z. and A.R.M.; data curation, C.L.J., A.R.M., T.C., V.N., N.S.; writing—original draft preparation, C.L.J., A.R.M., M.E.M., J.N.C., and Z.S.H.; writing—review and editing, K.P., S.J.K., A.A.K., P.B.N., V.N., N.S., S.C., and Z.S.H.; All authors have read and agreed to the published version of the manuscript.

**Funding:** This research was funded by USAID Wonders of the Mekong Cooperative Agreement (AID-OAA-A-00057 to Z.H. and S.C.). C.J. was also partially funded by NASA (NNX14AR62A), BOEM (MC15AC00006), and NOAA's support of the Santa Barbara Channel Marine Biodiversity Observation Network.

**Institutional Review Board Statement:** Not applicable.

**Informed Consent Statement:** Not applicable.

**Data Availability Statement:** All data is provided in Supplementary Materials.

**Acknowledgments:** We are grateful to the MRC for kindly providing data used in the species list and all the researchers populating genetic reference databases with sequence information.

**Conflicts of Interest:** The authors declare no conflict of interest.

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
