# Peer review of "Are Genetic Reference Libraries Sufficient for Environmental DNA Metabarcoding of Mekong River Basin Fish?"

_water, doi:10.3390/w13131767_

Round 1

Reviewer 1 Report

Article review

Title: Are Genetic Reference Libraries Sufficient for Environmental DNA Metabarcoding of Mekong River Basin Fish?

Comments to the authors:

First of all, congratulations for the good effort on writing this manuscript. Overall, this manuscript is well-written with very minor grammatical error. The authors’ discussion on eDNA metabarcoading is convincing and organized. The authors successfully addressed and identified many of the eDNA metabarcoding knowledge gaps, hence, help to improve the reliability of eDNA metabarcoding applications in the Mekong River Basin. Most of the reference used are also up to date. The minor grammatical error mostly involves the need to add the article a/the/an, the authors are suggested to double check and consider choice of word that can be used to simplify the sentences. 

Some little comment:

- Please fill up the left side of the Title page!

- Please use space between Authors’ name and Affiliation numbers.

- Please use monogram of authors after the e-mail addresses.

- I think the 14 and 15 affiliation are same! Please delete all ORCID identifier.

- The Introduction and Materials and Methods chapters are mixed and too long, please revise these chapters.

- Figure 2. Please use bigger text!

- Figure 3. Please use bigger text in case of both axis and I think you have to use other scales on the Y axis to present better your results. And delete some symbols from the left side of Figure.

Author Response

First of all, congratulations for the good effort on writing this manuscript. Overall, this manuscript is well-written with very minor grammatical error. The authors’ discussion on eDNA metabarcoading is convincing and organized. The authors successfully addressed and identified many of the eDNA metabarcoding knowledge gaps, hence, help to improve the reliability of eDNA metabarcoding applications in the Mekong River Basin. Most of the reference used are also up to date. The minor grammatical error mostly involves the need to add the article a/the/an, the authors are suggested to double check and consider choice of word that can be used to simplify the sentences. 

R: We reduced the Introduction and Materials and Methods by 3% as measured by word count and checked the manuscript for typographical errors with attention to inserting appropriate articles.

Some little comment:

- Please fill up the left side of the Title page!

R: Thank you for your review. We have filled up all of the information that we have on the left side of the title page. Note the page range, DOI, and dates of accepted and published are to be determined (TBD)

- Please use space between Authors’ name and Affiliation numbers.

R: Spaces added.

- Please use monogram of authors after the e-mail addresses.

R: Monograms added.

- I think the 14 and 15 affiliation are same! Please delete all ORCID identifier.

R: ORCID removed and affiliations condensed

- The Introduction and Materials and Methods chapters are mixed and too long, please revise these chapters.

            R: We reduced the Introduction and Materials and Methods by 3% as measured by word count.  We retained the description of the GAPeDNA methodology with default results in the introduction as this baseline information is critical for communicating our research motivation for further refinement of species lists and using multiple primer pairs.

- Figure 2. Please use bigger text!

            R: We increased the size of the set labels (FCFB, GAP, MRC), but kept the percentages and counts small in order to accommodate small subsets to be seen.  Specifically, the font labeling the <1% (2) overlap of FCFB and GAP sets needs to be small.

- Figure 3. Please use bigger text in case of both axis and I think you have to use other scales on the Y axis to present better your results. And delete some symbols from the left side of Figure.

R: We increased the font size of the X and Y axis labels.  It is unclear what symbols the are being referring to. We changed the panel lettering to be consistent with Water directions (a), (b), (c), (d), and (e). We chose to keep the y-axis scale the same across all sets to make them comparable to each other. 

Reviewer 2 Report

It is a very well written and detailed manuscript. It covers all aspects of a quality article; however, it has a few extensive sections (e.g., Introduction, Materials and Methods) making the reader lost at times. Readability can be improved if the authors can condense the contents by eliminating some minor details.

Another minor change is required at Lines 223-228: please provide R reference and EulerR package reference.

Author Response

It is a very well written and detailed manuscript. It covers all aspects of a quality article; however, it has a few extensive sections (e.g., Introduction, Materials and Methods) making the reader lost at times. Readability can be improved if the authors can condense the contents by eliminating some minor details.

R: Thank you for your review. We have attempted to clarify throughout with emphasis on the introduction and Materials and Methods and taken extra effort to condense contents and eliminating minor details. 

Another minor change is required at Lines 223-228: please provide R reference and EulerR package reference.

R: Reference added. 

Reviewer 3 Report

The authors aim to summarize and carefully analyse all available information in the databases for fish identification via eDNA metabarcoding in MRB. The combined species list created with critical and up-to-date methods and resulted a more comprehensive, relaibale dataset which might be very important for the future preservation programs. The results are well presented and among others  the conclusions include relevant informations on the usefullness  and potenital role of the primers included in the analysis.

Author Response

The authors aim to summarize and carefully analyze all available information in the databases for fish identification via eDNA metabarcoding in MRB. The combined species list created with critical and up-to-date methods and resulted a more comprehensive, reliable dataset which might be very important for the future preservation programs. The results are well presented and among others the conclusions include relevant information on the usefulness  and potential role of the primers included in the analysis.

R: Thank you for your review.